# *BmSuc1* Affects Silk Properties by Acting on Sericin1 in *Bombyx mori*

**DOI:** 10.3390/ijms23179891

**Published:** 2022-08-31

**Authors:** Liangli Yang, Yue Zhao, Quan Gan, Dan Liang, Rui Shu, Song Jiang, Ruiping Xie, Yan Meng

**Affiliations:** 1School of Life Sciences, Anhui Agricultural University, 130 West Changjiang Road, Hefei 230036, China; 2Anhui Academy of Agricultural Sciences, 40 Nongke South Road, Hefei 230036, China; 3Anhui International Joint Research and Development Center of Sericulture Resources Utilization, 130 West Changjiang Road, Hefei 230036, China

**Keywords:** *Bombyx mori*, BmSUC1, Ser1, CRISPR/Cas9, mechanical properties

## Abstract

*BmSuc1*, a novel animal-type β-fructofuranosidase (β-FFase, EC 3.2.1.26) encoding gene, was cloned and identified for the first time in the silkworm, *Bombyx mori*. *BmSuc1* was specifically and highly expressed in the midgut and silk gland of *Bombyx mori*. Until now, the function of *BmSuc1* in the silk gland was unclear. In this study, it was found that the expression changes of *BmSuc1* in the fifth instar silk gland were consistent with the growth rate of the silk gland. Next, with the aid of the CRISPR/Cas9 system, the *BmSuc1* locus was genetically mutated, and homozygous mutant silkworm strains with truncated β-FFase (BmSUC1) proteins were established. *BmSuc1* mutant larvae exhibited stunted growth and decreased body weight. Interestingly, the molecular weight of part of Sericin1 (Ser1) in the silk gland of the mutant silkworms was reduced. The knockout of *BmSuc1* reduced the sericin content in the silkworm cocoon shell, and the mechanical properties of the mutant line silk fibers were also negatively affected. These results reveal that BmSUC1 is involved in the synthesis of Ser1 protein in silk glands and helps to maintain the homeostasis of silk protein content in silk fibers and the mechanical properties of silk fibers, laying a foundation for the study of BmSUC1 regulation of silk protein synthesis in silk glands.

## 1. Introduction

Sucrose is one of the main products of photosynthesis and the most common transported sugar in plants, and it is also an easily assimilated macronutrient that provides a carbon or energy source for insects to fulfil the requirement of physiological metabolism [1,2]. Sucrose can be hydrolysed by α-glucosidase (EC 3.2.1.20) acts on α-glucosyl residue and by β-fructofuranosidase (β-FFase, EC 3.2.1.26) acts on β-fructosyl residue. α-glucosidase belongs to the hydrolase GH13 family and is widely found in animals, plants, and microorganisms [3]. β-FFase, which belongs to the glycoside hydrolase GH32 family, is abundant in microorganisms and plants and can release monosaccharide units by hydrolysis of sucrose [4]. Moreover, the β-FFase also has transfructosylation activity, which can convert sucrose into fructooligosaccharides (FOS) [5,6]. However, there are few reports on animal β-FFase.

Insects can utilize plant-derived sucrose as a source of food nutrition. However, insect sucrase activity is generally thought to depend mainly on α-glucosidase [1,7]. As the research continued, the researchers discovered the presence of β-FFase activity in insects and speculated that β-FFase is not synthesized by intestinal bacteria [8,9]. In 2008, a novel gene (named *BmSuc1*) of β-FFase was cloned and identified in the silkworm *Bombyx mori*. This provided the first direct evidence for the existence of β-FFase in an animal genome [10]. Subsequently, the β-FFase gene was successively discovered in the larvae of *Helicoverpa armigera* [11], *Manduca sexta* [12], *Dendroctonus ponderosae* [13], *Sphenophorus levis* [14], and *Agrilus planipennis* [15]. The insect β-FFase gene is thought to have been acquired via horizontal gene transfer from bacteria, including the origin of the silkworm gene *BmSuc1* [10,12,13]. The BmSuc1 gene was highly expressed in the midgut and in the anterior and middle parts of the silk gland, and *BmSuc1* encodes a functional β-fructofuranosidase, whose enzymatic activity was not inhibited by 1-deoxynojirimycin (DNJ) [2,10]. Our previous study reported that BmSUC1 acted as an essential sucrase by directly modulating the degree of sucrose hydrolysis in the silkworm larval midgut. Silencing *BmSuc1* significantly reduced glucose, activated maltase and trehalase in the midgut, and decreased glycogen and trehalose in the fat body, which resulted in larval malnutrition and abnormal petite phenotypes [2]. These results strongly support the hypothesis that silkworms can evade the toxic effects of mulberry leaf alkaloids by high expression of β-FFase in the midgut, then hydrolyse sucrose to absorb sugar nutrition [10].

The silk gland is vitally important for *Bombyx mori*, as it biosynthesizes the silk protein and spins the silk fiber. The silkworm synthesizes fibroin and sericin proteins in the silk gland to cocoon and pupates inside that shelter at the mature stage [16]. The silk gland is located on the left and right sides of the digestive tract and consists of the anterior silk gland (ASG), middle silk gland (MSG), and posterior silk gland (PSG). Sericin is mainly synthesized in MSG cells and silk fibroin is mainly synthesized in PSG cells [17]. In addition to silk proteins, numerous functional proteins exist in silk glands. Through comparative proteomics, Li et al. found that some proteins highly expressed in the anterior part of the MSG are associated with silk gland development and silk protein protection [18]. BmSerpin-16 is specifically and highly expressed in ASG and MSG of *Bombyx mori*, which is important for maintaining the stability of the silk secretion environment [19]. BmOsiris9a is specifically expressed in silk glands and exists in cocoon silk, and the overexpression of BmOsiris9a can improve the mechanical properties of silk fibers [20,21]. For *BmSuc1*, which is also highly expressed in silk glands, previous studies have already been carried out. For example, through the proteomic analysis of the silk glands of silkworms, Dong et al. found that the expression of β-FFase gradually decreased from the 5th instar to the wandering stage and speculated that β-FFase may be involved in the glycosylation of flavonoids [22]. Subsequently, Guo et al. detected the presence of β-FFase in cocoon silk, indicating that β-FFase is also a component of cocoon silk protein [23]. However, it is not clear whether BmSUC1 is related to the development of silk glands and the synthesis of silk protein.

In the present study, we investigated the relationship between silk gland growth rates and *BmSuc1* expression patterns. The *BmSuc1* mutant silkworm was obtained using the Cas9/small guide RNA (sgRNA) system, and the changes of silk proteins in silk gland tissues before and after *BmSuc1* deletion were analyzed. Finally, we also investigated the effect of *BmSuc1* deletion on the mechanical properties of silk fibers. Our findings lay the foundation for the exploration of the functions of specifically expressed genes in silk glands.

## 2. Results

### 2.1. Expression Pattern of BmSuc1 Is Identical to the Growth Rate of Silk Gland

The development of silk glands and the synthesis of silk proteins in *Bombyx mori* are mainly completed at the fifth instar [24,25]. To determine the relationship between *BmSuc1* expression and silk gland development, we first recorded the body weight of silkworm larvae from the fifth instar to the wandering stage and silk gland weights in these larvae to investigate the developmental pattern of silk glands throughout the fifth instar. It was found that the body weight of silkworms increased rapidly from the early fifth instar to the end of the fifth instar (Figure 1A). With the increase of silkworm body weight, the weight of the silk glands increased rapidly and reached the maximum value in the wandering stage (Figure 1B), and the proportion of silk glands in individuals gradually increased (Figure 1C). The growth rate of silk glands was the fastest in the early fifth instar, then gradually slowed down (Figure 1D). A previous study found that *BmSuc1* was highly expressed in ASG and MSG of the day 3 fifth instar larvae of *Bombyx mori* [2]. We investigated the mRNA levels of *BmSuc1* in the ASG and MSG tissues of the whole fifth instar silkworm by RT-PCR and qRT-PCR. It was found that *BmSuc1* was highly expressed in the ASG and MSG of the fifth instar silkworm, with the highest expression in the early fifth instar, and gradually decreased over time (Figure 2A,B). At the same time, the expression pattern of BmSUC1 protein in fifth instar silkworms was analyzed by SDS-PAGE and western blot. The results showed that the protein abundance in the silk glandular of *Bombyx mori* was much higher than that in the silk gland luminal (Figure 2C,E). For the BmSUC1 protein, its expression changes were the same as the mRNA transcription level, and it was shown that the expression level in the luminal was higher than that in the glandular (Figure 2D,F). These results indicate that the expression pattern of *BmSuc1* (Figure 2B) is consistent with the growth rate of silk glands (Figure 1D), suggesting that BmSUC1 in silk glands may be related to silk gland development.

### 2.2. Body Weight Decreased in BmSuc1 Knockout Mutant Larvae

To investigate the function of *BmSuc1* in the silk gland of silkworms, we genetically ablated *BmSuc1* using a transposon-based, Cas9/sgRNA-mediated mutagenesis system [26,27]. Two independent transgenic lines were established by transposon-mediated germline transformation. One transgenic line expressed two sequence-specific sgRNAs against *BmSuc1* (Figure 3A) under the control of the *BmU6* promoter [28,29], while the other line expressed Cas9 under the control of the germ-cell-specific promoter *Bmnos* [27]. Each line also expressed an IE1 promoter-derived fluorescent marker (DsRed2 in the sgRNA-expressing lines and EGFP in the Cas9-expressing lines) (Appendix A) to facilitate the screening of positive individuals from the embryonic stage [27]. In the F1 hybrid progeny of the sgRNA and Cas9 lines, positive individuals were screened for containing both red and green fluorescently tagged proteins (Figure 3B), and their somatic mutagenesis was identified by PCR-based analysis (Appendix A) and subsequent sequencing (Appendix A), indicating that successful mutagenesis was induced by the transgenic CRISPR/Cas9 system. Compared with the wild-type (WT) animals, the knockout of *BmSuc1* did not interfere with silkworm fertility. To obtain heritable, non-transgenic, homologous mutants, two adults with complete deletion events were selected based on PCR identification and sequencing results of F2 generation adults with fluorescent deletion (Appendix A), then backcrossed with WT moths, followed by a series of hybridization strategies and PCR-based screening experiments (Appendix A) as previously described [26]. Ultimately, two independent *BmSuc1*-deficient homozygous lines were established (Figure 3C and Appendix A).

One mutant line (Δ*BmSuc1*-L1) had a 1265-bp genomic DNA deletion, resulting in a 1265-bp deletion in the ORF and yielded a truncated 69-aa protein that was 419-aa shorter than the WT BmSUC1 protein (Figure 4A,B). Another mutant line (Δ*BmSuc1*-L2) has a 1232-bp genomic DNA deletion, resulting in a 1232-bp deletion in the ORF and yielded a truncated 80-aa protein that was 408-aa shorter than the WT BmSUC1 protein (Figure 4A,B). Furthermore, compared with WT, *BmSuc1* mutant larvae extended the duration of the final larval stage by approximately 12 h, and the individual weight of the mutant was lower than that of the WT at the early fifth instar (Figure 4C). It indicated that the knockout of *BmSuc1* had a negative impact on the growth and development of silkworm larvae, which were consistent with previous reports [2].

### 2.3. Sucrose Hydrolase Activity Lost in Silk Gland of BmSuc1 Mutant

Subsequently, we investigated the sucrose hydrolase activity of ΔBmSUC1 in vitro and in vivo. We first synthesized and purified two truncated BmSUC1 and WT BmSUC1 proteins in vitro using a prokaryotic expression system (Appendix A). The purified recombinant protein was confirmed by western blot with different antibodies (Figure 5A). The level of sucrose hydrolase activity of the recombinant protein was detected by DNS (dinitrosalicylic acid) method, and it was found that both ΔBmSUC1 had lost the enzymatic activity of hydrolyzing sucrose (Figure 5B). Next, in order to explore the changes in the sucrose hydrolysis activity of BmSUC1 in the mutant strain silkworm, and the role of BmSUC1 in the sucrose hydrolysis of silk gland tissue of the silkworm larvae, we explored the catalytic ability of sucrase in ASG and MSG total proteins (distinguishing glandular and luminal) of the fifth instar silkworm by adding DNJ, an inhibitor of α-glucosidase, to the reaction mix (Figure 5C). When the DNJ was added to the reaction, the sucrose hydrolase activity in the silk gland of the WT strain did not change significantly. Compared with the WT line, the sucrose hydrolase activity in the silk glands of the mutant larval was almost lost, which was consistent with the loss of sucrose hydrolase activity in the purified recombinant protein ΔBmSUC1 in vitro (Figure 5B), indicating that ΔBmSUC1 in the mutant silkworm had been lost the typical β-FFase character.

### 2.4. Molecular Weight of Ser1 Decreased in BmSuc1 Mutant

To explore the effect of knocking out BmSUC1 on silk protein synthesis in silkworm silk glands, we performed SDS-PAGE and western blot analysis on total proteins in silk gland tissues from different parts of the silkworm at day 3 fifth instar and wandering stage, respectively. The results showed that the total proteins of ASG and MSG glandular and PSG were not significantly different in both mutant and WT silkworms. Interestingly, in the ASG and MSG luminal of the wandering stage silkworm, compared to the WT, there is a protein band in mutant L1 that becomes lighter in color, and there is an additional protein band of smaller molecular weight below it (indicated by red arrows; Figure 6A). Subsequently, we examined the expression of tissue-specific silk proteins Ser1, BmOsiris9a, and fibrohexamerin/P25 fibroin in silk glands. The results showed that the expression levels of BmOsiris9a and P25 were not significantly different between mutant and WT silkworms. BmOsiris9a was only detected in ASG and MSG, but not in PSG; P25 was detected in both ASG and MSG and PSG (Figure 6B), which is consistent with previous results [17,21]. Interestingly, in ASG and MSG, the BmSer1 antibody detected only one product signal band in WT silkworm, whereas two product signal bands were detected in the mutant line, and the product signal was stronger in the wandering stage (indicated by a blue arrow; Figure 6B). Based on the above results, we speculate that the smaller molecular weight band in the mutant L1 in the SDS-PAGE results is likely to be Ser1.

To further confirm our speculation, we performed native PAGE separation of total proteins in the luminal of ASG and MSG in the wandering stage of the mutant L1 and WT silkworms. In the WT silkworm, two closely spaced bands were marked as differential band 1, and two protein bands appeared at the corresponding positions in the mutant strain, marked as differential band 2 and differential band 3 (Figure 7A). The results of mass spectrometry analysis of the three differential bands found that Ser1 components and various other protein components were contained in all three samples (Figure 7B), indicating that the three differential protein bands were protein complexes. Combined with the Western blot results (Figure 6B), it was indicated that the knockout of *BmSuc1* resulted in the decrease in molecular weight of some Ser1 proteins.

### 2.5. Knockout of BmSuc1 Reduces the Strength of Silk Fibres

Cocoons of mutant and WT lines were harvested, and raw silk was coiled from the cocoons. To characterize the effect of knocking out *BmSuc1* on sericin content, sericin in cocoon shells was extracted and weighed, and it was found that sericin content in mutant L1 was decreased (Figure 8A). To determine the effect of *BmSuc1* knockout on silk fiber mechanical properties, mechanical tests were performed using raw silk fibers. The results showed that the breaking elongation and elongation at the break of the mutant silk were higher than those of the WT silk. The breaking force and breaking strength of WT filaments were greater than those of mutant tethered filaments (Figure 8B). It is speculated that the knockout of *BmSuc1* reduces the molecular weight of part of Ser1, resulting in a decrease in the overall quality of sericin, which in turn negatively affects the mechanical properties of silk fibers.

## 3. Discussion

As the first animal-type β-FFase gene to be cloned and identified, *BmSuc1* is specifically and highly expressed in the midgut and silk gland tissues of silkworms [2,10]. BmSUC1 plays an important role in the metabolism of sugar nutrition in the midgut of silkworms and has helped silkworm larvae to break into a new and essential path to absorb sugar nutrients through alkaloid sugar mimics in the mulberry leaves [2]. However, the function of *BmSuc1* in silkworm silk glands is poorly understood. To broaden our understanding of *BmSuc1* function in silk gland tissue of *Bombyx mori*, we genetically abolished *BmSuc1* by Cas9/sgRNA-mediated targeted mutagenesis, revealing that BmSUC1 is involved in the synthesis of sericin 1 protein in the silk gland and plays an important role in maintaining the excellent mechanical properties of silk fibers.

The silk gland of silkworms is a specialized tissue; morphologically and functionally, it can be divided into ASG, MSG for synthesizing and secreting sericin, and PSG for synthesizing and secreting fibroin [30]. The silk glands of the fifth instar larvae developed rapidly, accounting for 30.1% of the body weight of larvae (Figure 1C), which corresponds to the previous findings that the fifth instar is an important period for silk gland development and silk protein synthesis [24,25]. The research on silk glands is relatively clear about the expression and transcriptional regulation of silk protein genes, the development of silk glands, and the synthesis of silk proteins; for example, the transcription factor Bmsage participates in the regulation of silk fibroin heavy chain genes through the interaction with SGF1 [31] and the discovery and characterization of the silk protein constituent BmOsiris9a in the silk gland [21]. However, studies on functional proteins in silk glands are still very scarce. To understand the function of *BmSuc1* in silk glands, we first investigated the expression of *BmSuc1* in ASG and MSG from the transcriptional and protein levels. The time expression profiling results showed that *BmSuc1* was highly expressed in the early fifth instar of *Bombyx mori* (Figure 2), which was consistent with the higher growth rate of silk glands in the early fifth instar (Figure 1D). We speculated that *BmSuc1* might be involved in the development of silk glands.

The amino acid sequence alignment results (Figure 4A) showed that BmSUC1 in the CRISPR/Cas9-*BmSuc1* mutant lines lacked most of the amino acids, including three conserved sequences and three restriction sites [32,33]. It is speculated that the β-FFase function has been lost in the mutant line, which was also confirmed by in vivo and in vitro sucrose hydrolase activity experiments (Figure 5 and Appendix A). It is well known that sucrose can be hydrolyzed by α-glucosidase and β-FFase. When the BmSUC1 activity was lost in the midgut of mutant larvae, α-glucosidase activity was increased (Appendix A). However, even with α-glucosidase as compensation, the mutant larvae still showed developmental retardation and lighter weight (Figure 4C). The reduction of mutant larvae weight is most likely due to the loss of BmSUC1 sucrose hydrolase function. This result further confirms the previous findings that the sugar metabolism of silkworms is a complex process, in which BmSUC1 plays a major role [2].

The silk gland of silkworms can be divided into two parts in space: silk glandular and silk gland luminal. Silk protein is mainly synthesized in the glandular and then secreted into the luminal for transportation and storage [30]. The expression of BmSUC1 in the luminal of the silk gland is higher than that in the glandular (Figure 2D,F), and Guo et al. also detected the presence of BmSUC1 protein in cocoon silk [23], indicating that silkworm BmSUC1 is synthesized in the glandular and secreted into the luminal to directly interact with silk protein to play its function. The findings of Dai et al. suggest that BmSUC1 has a role in maintaining silk gland metabolic homeostasis [34]. Loss of BmSUC1 resulted in a disturbance of the silk secretion environment, resulting in a decrease in the secretion of sericin protein and a decrease in the content of sericin in the cocoons (Figure 8). However, by using an anti-Ser1 antibody, we found no significant changes in Ser1 content in the BmSuc1-L1 mutant (Figure 6). It is well known that sericin protein is a collective term for several proteins (17), and the constant content of one sericin does not contradict the decrease of the total sericin content (Figure 8). Interestingly, the molecular weight of part of Ser1 protein was reduced in the ASG and MSG of mutants in our study (Figure 6 and Figure 7). It has been reported that Ser1 has multiple splicing isoforms and is different among strains [35]. In this study, one Ser1 band was mainly detected in the WT strain, so the two bands detected in the mutant strain may represent different isomers or dimers of Ser1 protein. β-FFase, also known as invertase, can transfer fructosyl to sucrose molecules to form FOS, such as kestose [36,37]. Studies have confirmed that BmSUC1 in silkworms has both sucrose hydrolase and glycosyltransferase activities [2,32]. The activity of sucrose hydrolase in WT silk glands did not change with the addition of DNJ, indicating that only β-FFase, but not α-glucosidase, was present in silk gland tissue (Figure 5C). Compared with the sucrose hydrolase function of BmSUC1 in the midgut, BmSUC1 may mainly function as a glycosyltransferase in the silk gland. Glycosyltransferases play an important role in the process of protein glycosylation. Therefore, we speculate that the post-translational modification of the mutant silkworm Ser1 has changed. In particular, glycosylation modifications are more common in silk fibroin. For example, both silk fibroin P25 and Fib-L proteins have glycosylation modifications and lead to protein bands of different molecular weights [38]. However, the specific process of BmSUC1 involved in Ser1 protein synthesis and possibly involved in Ser1 post-translational modification processing in silkworms remains unclear and awaits further study.

Cocoon silk is mainly composed of fibroin and sericin proteins, and the composition of the protein determines the performance of the silk [30]. For example, by overexpressing BmOsiris9a in the silk gland, Cheng et al. found that the crystallinity of cocoon silk was reduced and the mechanical properties of silk fibers were better [39]. In the present study, deletion of *BmSuc1* caused a decrease in sericin content in the cocoon shell and a decrease in the breaking strength of raw silk (Figure 8). It is generally believed that fibroin plays a dominant role in the mechanical properties of silk, while the contribution of sericin to the mechanical properties is generally negligible [40]. However, sericin in raw silk has a certain contribution to the mechanical properties of raw silk, and it was found that the breaking strength decreased by 20% when sericin was completely removed [41]. Studies have confirmed that sericin has low crystallinity, high fluidity, and high hydrophilicity. After the sericin is completely removed, the mechanical properties of the raw silk decrease. When a certain amount of sericin is retained in the raw silk, the sericin acts as a matrix in it, which can be regarded as a kind of fiber with multiple silk fibroin fibers as reinforcement materials [42]. The results of this study indicated that the deletion of *BmSuc1* resulted in the reduction of sericin content in silk fibers, which in turn negatively affected the mechanical properties of silk fibers.

In conclusion, BmSUC1 contributes to the normal synthesis of Ser1 protein in silk glands and plays an important role in maintaining the homeostasis of silk protein content in silk fibers and the mechanical properties of silk fibers. However, it is still unclear whether the loss of BmSUC1, as we have speculated, hinders the glycosylation modification of the Ser1 protein, resulting in the reduction of the molecular weight of part of Ser1, which requires further experiments to determine. Therefore, in the following research, we will have a deeper understanding of the relationship between BmSUC1 and Ser1, hoping to further reveal the molecular regulation mechanism of BmSUC1 acting on Ser1.

## 4. Materials and Methods

### 4.1. Silkworms and Vectors

The multivoltine and monophagous *B. mori* strain, Nistari, was used as the basic transgenic strain. Silkworm larvae were fed fresh mulberry leaves and maintained at 25 °C under standard conditions [43]. The pMD19T plasmid was purchased from TaKaRa (Dalian, China). The PUC57 plasmid was purchased from TSINGKE Biological Technology (Nanjing, China). The pET32a plasmid, pET24b-*BmSuc1* recombinant plasmid, and anti-BmSUC1 polyclonal antibody were maintained in our laboratory. The plasmid PXL-IE1-DsRed2-U6-U6 (*Sal* I, *Nhe* I), containing the *B. mori* U6 promoter, was used to construct the sgRNA expression vector.

### 4.2. RNA Isolation, cDNA Synthesis, and PCR

Total RNA was extracted from silk gland tissues on the fifth instar larval using TRIzol reagent (Sangon, Shanghai, China) according to the manufacturer’s instructions. Subsequently, the RNA was treated with DNase I (TaKaRa, Dalian, China) to remove genomic DNA. For reverse transcription, the ReverAid First Strand cDNA Synthesis Kit (Sangon, Shanghai, China) was used with 1 μg of total RNA according to the manufacturer’s instructions. Reverse transcription PCR (RT-PCR) and quantitative real-time PCR (qRT-PCR) methods were performed to analyse the transcription level of *BmSuc1* in silk gland tissues of silkworm larvae. The cycling conditions of RT-PCR were 95 °C for 5 min, followed by 29 cycles of 95 °C for 10 s, 51 °C for 20 s, and 72 °C for 20 s. PCR products were separated by 1.0% agarose gel electrophoresis. qRT-PCR was conducted using the CFX96 system (Bio-Rad, Hercules, CA, USA) and SYBR green PCR master mix (Bio-Rad, Hercules, CA, USA). Each qRT-PCR was performed under the following conditions: denaturation at 95 °C for 5 min, followed by 40 cycles at 95 °C for 10 s, 51 °C for 15 s, and 72 °C for 20 s. The 2^−∆∆Ct^ method [44] was used to calculate the relative expression level of the target gene. *B. mori BmActin3* was used as an internal control. The primers used in RT-PCR and qRT-PCR are listed in Appendix A (Invitrogen, Shanghai, China). All experiments were performed in triplicate and repeated three times independently.

### 4.3. Western Blot

Total protein was extracted from different tissues of larvae using the Tissue or Cell Total Protein Extraction Kit (Sangon, Shanghai, China) [2]. Extracted total protein or purified recombinant protein were separated, transferred, and analyzed by immunoblotting as described in previous reports [2,45]. The anti-β-Actin or the anti-BmSUC1 rabbit polyclonal antibodies prepared in our previous study [32] was diluted to 1:2500 or 1:500 and in 5% (*v*/*v*) skimmed milk in PBST, and the secondary antibody of goat anti-rabbit IgG conjugated with HRP (Sangon, Shanghai, China) were diluted 1:5000 in the same blocking solution. The final detection was performed by using HRP-DAB Horseradish Peroxidase Color Development Kit (Sangon, Shanghai, China). Other primary antibodies used in this study included anti-BmSer1 (1:10,000), anti-BmOsiri9a (1:10,000), and anti-BmP25 (1:5000) [21,39,46].

### 4.4. Investigation of the Developmental Pattern of the Silk Gland in Wild-Type Silkworm Larvae

To evaluate the developmental pattern of the silk gland in silkworm larvae, 3 individuals of the Nistari strain were randomly selected every day (24 h) from the newly molted fifth instar to the wandering stage. The total larval bodies were first weighed, then the silk glands isolated from 3 larvae were weighed. The larvae were fed fresh mulberry leaves in large quantities every day during this period. Daily total larval body or silk gland weights were averaged and compared.

### 4.5. Plasmid Construction

According to the target design principle of the CRISPR/Cas9 system GN19 + NGG (protospacer adjacent motif (PAM) recognition site, not involved in constituting sgRNA sequence) [47], two sgRNAs (siRNAs) were designed based on the open reading frame (ORF) of *BmSuc1* (GenBank: AB366559.1) on an online prediction website (www.crispr.dbcls.jp accessed on 28 July 2019). The sequences are shown in Appendix A. The sgRNA was synthesized by TSINGKE Biological Technology (Nanjing, China), and the sgRNA product was cloned into PUC57 according to the instructions. Next, the sgRNA1 fragment was digested with *Sal* I restriction endonuclease and ligated into PXL-IE1-DsRed2-U6-U6 (*Sal* I digested) vector by T4 ligase at 16 °C for 2 h, and the recombinant plasmid was named PXL-IE1-DsRed2-U6-sgRNA1-U6. The sgRNA2 fragment was then digested with *Nhe* I restriction endonuclease and ligated into the PXL-IE1-DsRed2-U6-sgRNA1-U6 (*Nhe* I digested) vector by T4 ligase at 16 °C for 2 h, and the recombinant plasmid was named PXL-IE1-DsRed2-U6-sgRNA1-U6-sgRNA2. The primers used are listed in Appendix A.

### 4.6. Silkworm Germline Transformation and Mutagenesis Analysis

For silkworm germline transformation, G0 preblastodermal Nistari embryos were injected with a mixture of transformation plasmids and helper plasmids, followed by incubation in a humidified chamber at 25 °C and 75% relative humidity for 10–12 days until larvae hatching [28,48]. Hatched larvae are raised to adults and sib-mated or backcrossed with WT adults. Next, G1 progeny was scored for the presence of the marker gene at the embryonic or newly-hatched silkworm stage under a fluorescence microscope (OLYMPUS DP72, Tokyo, Japan). For the transgenic CRISPR/Cas9 system, the Nos-Cas9 line (donated by Shanghai Institute of Plant Physiology and Biochemistry) was crossed with the U6-*BmSuc1*-sgRNA line as previously described [49]. Subsequently, genomic PCR and sequencing were performed using *BmSuc1* ORF primers to identify *BmSuc1* mutant alleles. The primers used are listed in Appendix A.

### 4.7. Screening Strategy and Establishment of Homozygous Mutant Strains and Phenotype Screening

To establish stable homozygous mutant lines, Nos-Cas9:U6-BmSuc-sgRNA (F1) somatic mutants were backcrossed to WT moths, and PCR was used to identify heterozygous F2 mutant animals (lacking fluorescence). The extraction method of animal genomic DNA refers to the previous method [26]. The selected F2 mutants were again backcrossed with WT moths, and the offspring of the cross would be approximately 50% heterozygotes and 50% WT animals. The F3 heterozygous animals were then sib-mated, and the offspring of the cross will yield approximately 25% WT, 50% heterozygous mutants, and 25% homozygous mutant animals. Finally, F4 homozygous mutants were then sib-mated to obtain 100% homozygous mutant animals for subsequent experiments. Details about the hybridization process are shown in Appendix A.

To investigate the growth and development of silkworm larvae, 20 individuals from the WT or CRISPR/Cas9-mediated mutants were randomly selected from a group (3 groups each) and fed abundantly with fresh mulberry leaves. At the fifth instar stage, each larva was weighed every 24 h intervals until the wandering stage. The individual weight of daily larvae was compared among strains.

### 4.8. Cloning, Expression, and Purification of Mutant and WT BmSUC1

DNAMAN software was used to predict the amino acids of the protein encoded by the mutant *BmSuc1* (Δ*BmSuc1*) gene. Genetyx_version 7 software (Version 7.03, Genetyx Corporation, Tokyo, Japan) was used to overlap amino acid sequences. To obtain the ΔBmSUC1 protein for biochemical analysis, the ORF of Δ*BmSuc1* was amplified by PCR, using specific primers designed according to the cDNA sequence of Δ*BmSuc1* and the mutant silk gland cDNA as the template. The PCR product was inserted into a pET32a-derived vector and transformed into *E. coli* BL21 (DE3) strain. Recombinant ΔBmSUC1 was expressed and purified in *E. coli* as described by Guo et al. [23]. The WT protein BmSUC1 was expressed using the pET24b-*BmSuc1* recombinant plasmid maintained in our laboratory [2,32] and was expressed and purified in the same manner as the mutant protein. The purity of the eluted protein was assessed by SDS-PAGE and detected by western blot using the anti-histidine antibody (1:2500), and the protein samples were stored at 80 °C until use. The primers used are shown in Appendix A.

### 4.9. Sucrase Activity Assays

The hydrolytic activity of sucrase on total protein or purified recombinant protein was determined according to the method of Daimon et al. [10] with some modifications, and the detailed reaction procedure was as previously described [2,32]. Reactions without enzyme addition served as controls. In addition, 0.25% DNJ (Santa Cruz Biotechnology, Santa Cruz, CA, USA) was added to the sucrose substrate to inhibit the activity of α-glucosidase in total protein. All experiments were performed independently in triplicate. One unit of enzyme is defined as the amount of enzyme per μg protein that catalyzes the production of 1 mmol glucose per minute (mmol/min/μg).

### 4.10. Native PAGE and Mass Spectrometry Analysis

Total ASG and MSG luminal proteins extracted from wandering stage silkworm larvae of mutant line 1 (L1) and WT lines were separated by 15% native PAGE. Then three differential protein bands (WT: 1; mutant L1: 2, 3) were excised from the native PAGE for mass spectrometry analysis in Beijing Huada Protein Research and Development Center Co., Ltd. (Huada, Beijing, China). Samples were digested by trypsin overnight at 37 °C and analyzed on the mass spectrometer (Q Exactive, Thermo-Fisher, Waltham, MA, USA). Raw data from mass spectrometry was used to identify peptides by searching against the silkworm proteins downloaded from the GenBank database using the software MasCot (version 2.2; Matrix Science, London, UK).

### 4.11. Extraction of Sericin from Silkworm Cocoon Shell

The cocoons used in this study were harvested from the rearing of mutant and WT silkworms and were kept at room temperature in a dry fume hood before use. To extract sericin, cocoon shells from WT or mutants were first washed with warm water to remove the spoils, dried at 60 °C (6–8 h), and cut into 1 cm^2^ pieces. Two grams of cocoon shell fragments and 200 mL of distilled water were added to a 125 °C-sterilization pot for 2.5 h to separate sericin. After filtering with gauze, the raw silk was washed with 100 mL of warm water 3 times, the washing solution was mixed with degumming liquid, and the solution was filtered with filter paper. The sericin solution was then rotary evaporated and concentrated in vacuo at 125 rpm, 60 °C, and a pressure of 0.1 MPa. The sample is concentrated to a volume of about 100 mL, pre-cooled (−80 °C, 2 h), and dried with a vacuum freeze dryer (sample temperature −33 °C, pressure 1 Pa) until it freezes into powder. The sericin content was calculated after weighing the powder. Three replicates were used for each silkworm line.

### 4.12. Mechanical Properties of Silk

The mutant silkworm and WT silkworm cocoons were reeled in the same way. The size of cocoon filament (WT: 1.381 deniers; mutant L1: 1.606 deniers) was determined by the fixed-length weighing method [50,51], and the number of cocoon filament silk fiber strands (WT: 15 strands; mutant L1: 13 strands) constituting the raw silk were determined according to the target cocoon filament size (about 21 D). The tensile sample tests were performed under ambient conditions using a constant rate elongation tester (CRE) with a gauge length of 50 mm and a constant speed of 150 mm/min of moving gripper movement. The strength reading accuracy was ≤0.01 kg (0.1 N), and the elongation reading accuracy was ≤0.1%. For each group, statistical analysis was performed using 20 ± 5 measurements with 3 replicates. Elongation at break, breaking force, and breaking strength were calculated.

### 4.13. Statistical Analysis

Data are presented as mean standard deviation (SD). Differences between groups were examined by a two-tailed Student t-test. Differences were considered statistically significant when the *p*-value was less than 0.05. Statistically significant differences were indicated with ‘*’ or ‘a, b, c’.

## Figures and Tables

**Figure 1 ijms-23-09891-f001:**
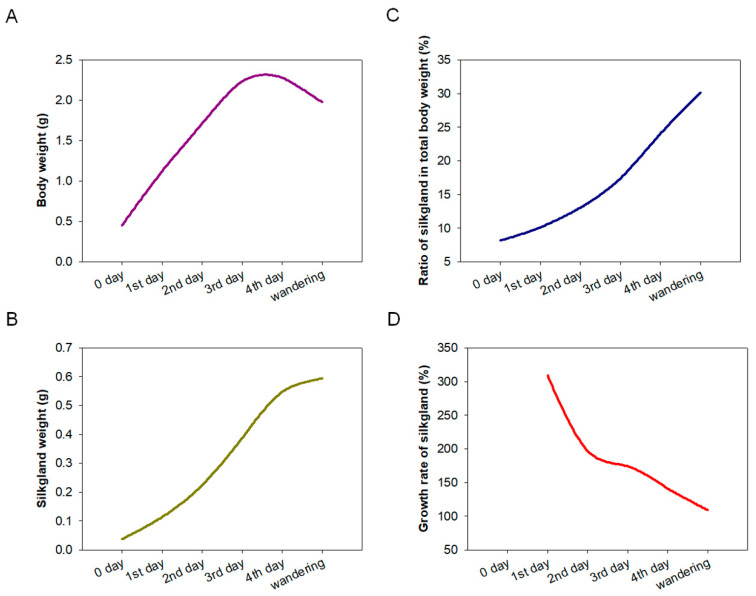
Investigation of the development of the silk gland during the fifth instar stage. The silk gland of the silkworm was dissected from the 0 days of the fifth instar (5L0D) to the wandering stage, average body and silk gland weight of 3 individuals per group at each stage. (**A**) Body weight, (**B**) silk gland weight, (**C**) ratio of silk gland to total body weight, and (**D**) growth rate of silk gland.

**Figure 2 ijms-23-09891-f002:**
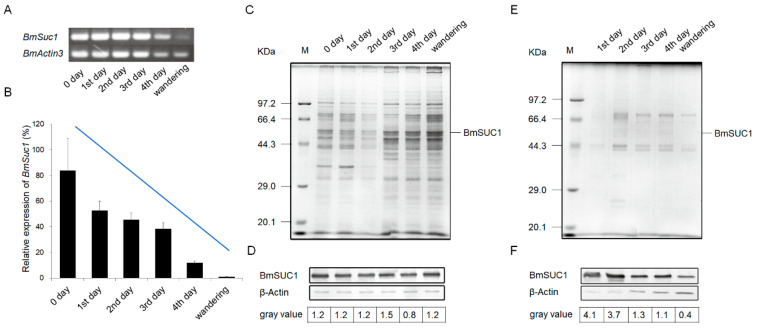
Expression profiles of *BmSuc1* in silkworm larvae anterior and middle parts of the silk gland during the fifth instar using (**A**) RT-PCR and (**B**) qRT-PCR. The total RNA was extracted from the 5L0D larvae to the wandering stage (expression level of *BmSuc1* at 5L0D is 100%). *BmActin3* was used as an internal control. Data in B represent mean ± SEM (*n* = 3). The SDS-PAGE and western blot analysis of (**C**,**D**) glandular and (**E**,**F**) lumina protein in anterior silk gland (ASG) and middle silk gland (MSG). (**C**,**E**) Total protein expression of ASG and MSG during the fifth instar by SDS-PAGE analysis. (**D**,**F**) Western blot analysis of ASG and MSG proteins obtained from fifth instar larvae using an anti-BmSUC1 antibody. β-Actin was used as the control. The amount of BmSUC1 is calculated by grayscale analysis.

**Figure 3 ijms-23-09891-f003:**
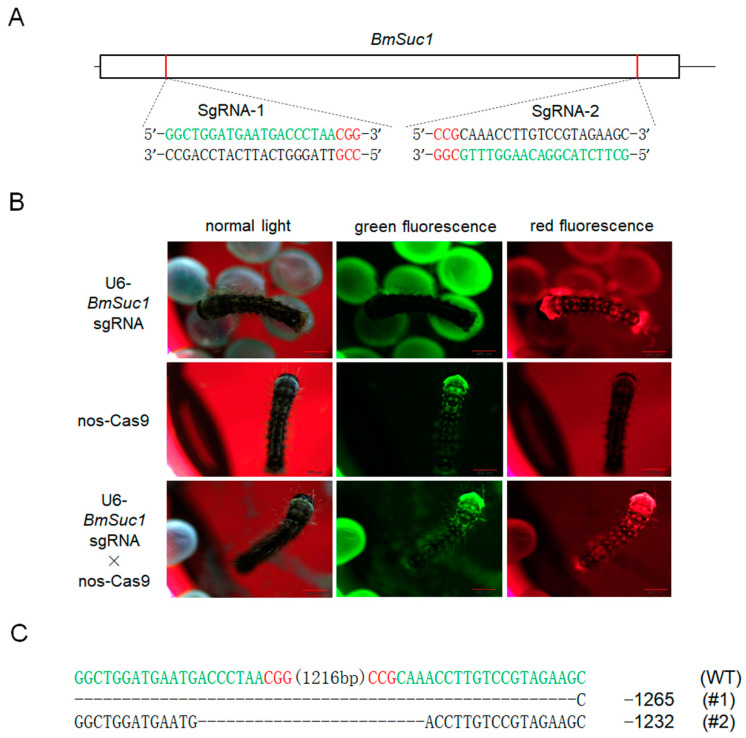
SgRNA target selection of the *BmSuc1* transgenic CRISPR/Cas9 system and fluorescence screening of transgenic silkworm and two selected mutation events. (**A**) Schematic diagram of the *BmSuc1* target sites. The black line represents the genome locus, the sgRNA-targeting sequence is in green, and the protospacer adjacent motif (PAM) sequence is in red. (**B**) Fluorescence screening of F1 transgenic silkworm. (**C**) The sequence shows the two selected mutation events. The sgRNA-targeting sequence is in green, and the PAM sequence is shown in red.

**Figure 4 ijms-23-09891-f004:**
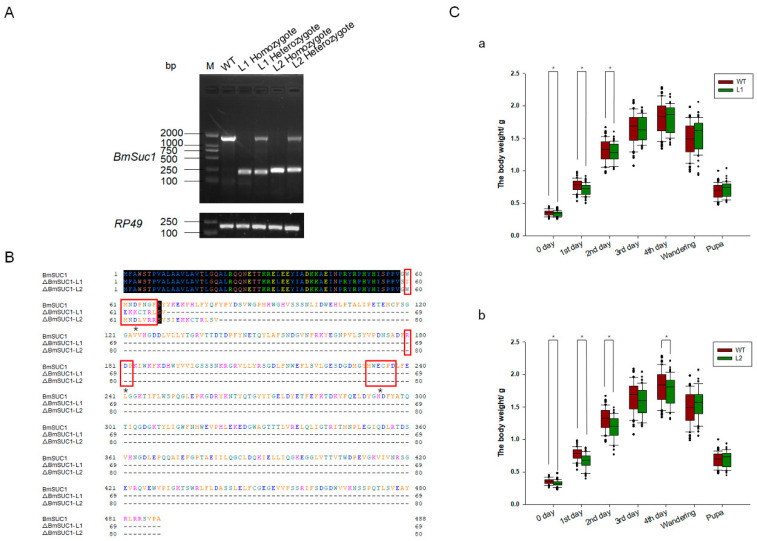
Comparison of *BmSuc1* among mutants and WT. (**A**) RT-PCR of *BmSuc1* of wild-type (WT), line 1 (L1) homozygote, L1 heterozygote, line 2 (L2) homozygote, and L2 heterozygote. Δ*BmSuc1* of mutant L1 had a 1265-bp deletion, and Δ*BmSuc1* of mutant L2 had a 1232-bp deletion in the ORF. (**B**) Amino acid sequence alignment of the BmSUC1 protein in WT, L1 homozygote, and L2 homozygote. ΔBmSUC1-L1 is 69 aa, and ΔBmSUC1-L2 is 80 aa. Red boxes represent three conserved domains (60–66, 180–182, 232–237). * Signify active sites (D63, D181, E234). (**C**) Effect of *BmSuc1* knockout on body weight development of silkworms. Comparison of body weight between (a) L1 mutant and WT and (b) L2 mutant and WT. The error bars represent SD; * *p* < 0.05.

**Figure 5 ijms-23-09891-f005:**
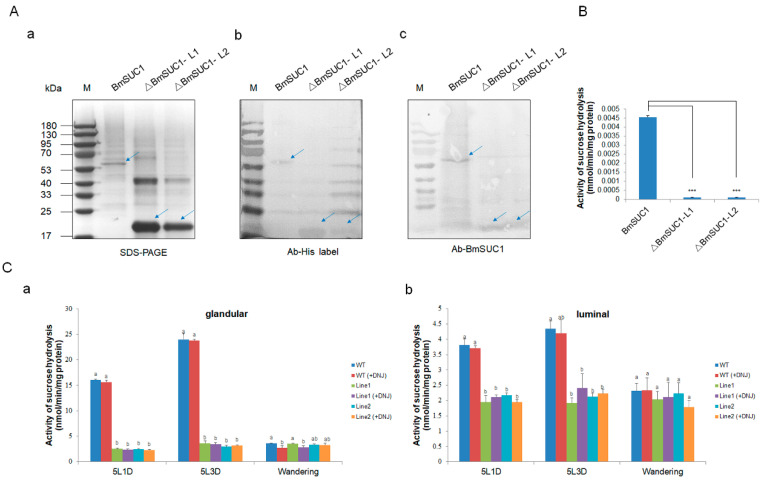
Determination of sucrose hydrolase activity. (**A**) Identification of three purified recombinant proteins by (a) SDS-PAGE, (b) western blot using anti-His labelled antibody, and (c) western blot using an anti-BmSUC1 antibody. (**B**) Sucrose hydrolase activity of ΔBmSUC1, which is expressed in prokaryotes. Error bars, SD; *** *p* < 0.001. (**C**) Determination of sucrose hydrolase activity in ASG and MSG of mutant and WT larva. Sucrose hydrolase activity (-DNJ) and β-fructofuranosidase activity (+DNJ) of (a) glandular proteins and (b) luminal proteins at the 5L1D, 5L3D, and wandering stages. Error bars, SD; *p*-values ≤ 0.05 are considered statistically significant (significant differences are represented by different letters a, b, and c).

**Figure 6 ijms-23-09891-f006:**
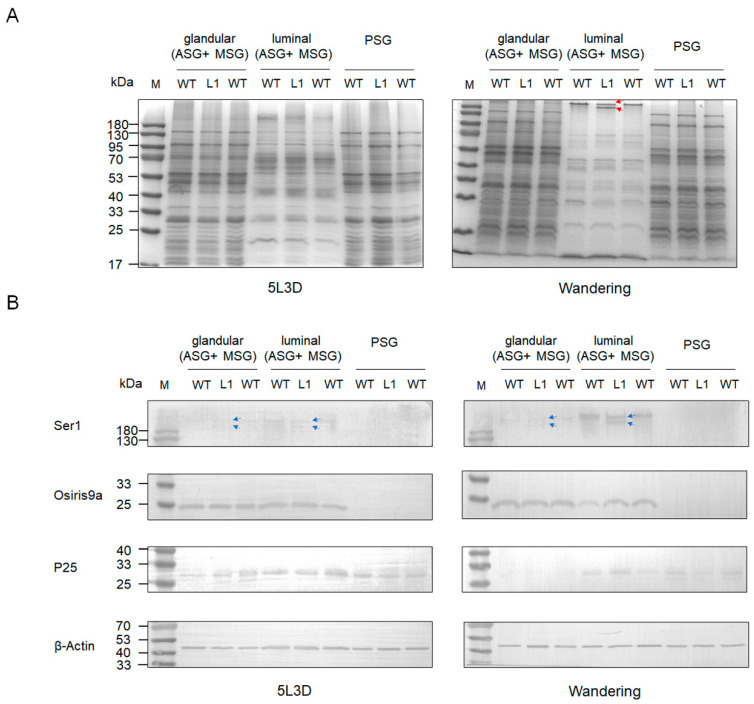
Protein analysis in different parts (glandular of ASG and MSG, luminal of ASG and MSG, and PSG) of larva silk gland tissue in different stages among mutant and WT by (**A**) SDS-PAGE. One protein band became lighter in color and a smaller molecular weight protein band appeared below it in the wandering stage of ASG and MSG luminal of mutant L1. The red lines represent the positions of light-colored protein bands and smaller molecular weight protein bands. (**B**) Western blot of glandular and luminal protein using anti-Ser1, anti-Osiris9a, and anti-P25 antibodies. β-Actin was used as the control. The blue lines indicate the position of the two Ser1 protein molecular weight bands that appear in the mutant L1.

**Figure 7 ijms-23-09891-f007:**
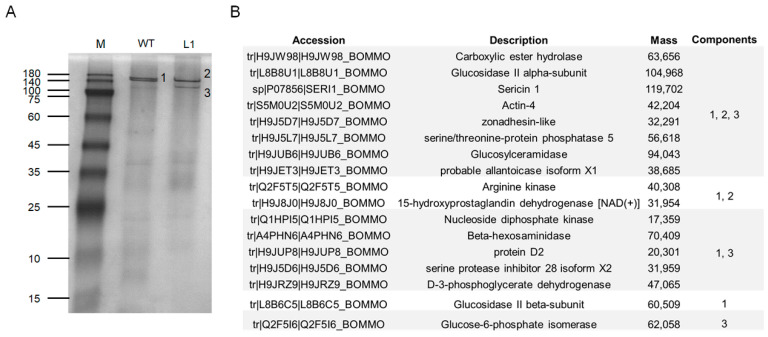
Mass spectrometry identification of differential components in the silk gland luminal (ASG and MSG) of wandering stage mutant and WT larva. (**A**) Total protein expression of ASG and MSG luminal in wandering stage by native PAGE analysis. (**B**) Summary of LC-MSMS analysis identification results for the 3 different protein bands (1, 2, and 3) obtained in (**A**).

**Figure 8 ijms-23-09891-f008:**
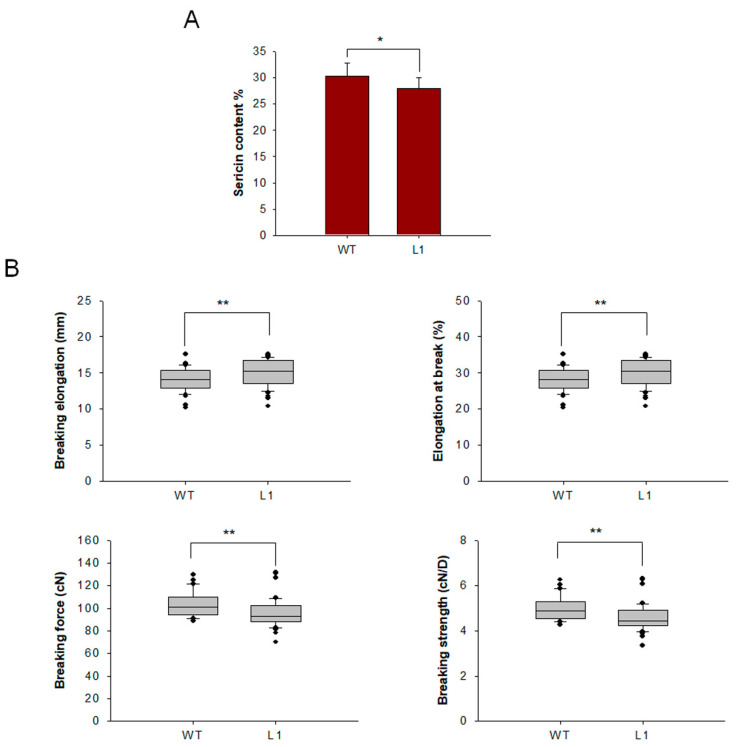
Comparison of sericin content and mechanical properties test of cocoon silks from mutant L1 and WT. (**A**) Sericin content in cocoons. (**B**) Three groups of measurements were made for each strain, 20 times for each group, and a total of 60 measurements were made. The loading distance of raw silk was 50 mm, and the tensile speed was 150 mm/min. Breaking elongation, Elongation at break, Breaking force and Breaking strength. Error bars, SD; * *p* < 0.05, ** *p* < 0.01.

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
