# Peer review of "BmSuc1* Affects Silk Properties by Acting on Sericin1 in *Bombyx mori"

_ijms, 2022, doi:10.3390/ijms23179891_

Round 1
Reviewer 1 Report
for comments see attached file

Reviewer 2 Report
This work uses homozygous mutant strains with truncated β-fructofuranosidase (BmSUC1) proteins obtained by CRISPR/Cas9 system to study the role of β-fructofuranosidase in silkworm. Of particular interest is the effect of one of the BmSuc1 mutations on the sericin composition of the silk gland lumen and on the sericin content of the cocoon, which is accompanied by a change in the mechanical properties of the silk fibers. This work contributes to the understanding of the mechanism of silk production in silkworms.
From the results of western blotting of glandular and luminal proteins using anti-Ser1 antibody (Figure 6), it is evident that the luminal sericin content is not reduced in the wandering stage of BmSUC1-L1 mutants, but only the amount of different isomers or dimers of Ser1 protein is changed. Subsequently, a reduction in cocoon sericin content was observed. How to explain this difference? Please discuss.
Histological sections of ASG, MSG and cocoon could reveal possible differences in the amount and distribution of sericin. The reduction in sericin content should be reflected in the thickness of the spun fibers. Has silk fiber thickness been compared in mutants and WT?
